# Ex Vivo Analysis of the Association of GFP-Expressing *L. aethiopica* and *L. mexicana* with Human Peripheral Blood-Derived (PBD) Leukocytes over 24 Hours

**DOI:** 10.3390/microorganisms12091909

**Published:** 2024-09-19

**Authors:** Medhavi Ranatunga, Andrew Deacon, Laurence S. Harbige, Paul Dyer, Joshua Boateng, Giulia T. M. Getti

**Affiliations:** 1Faculty of Science and Engineering, The University of Greenwich at Medway, Central Avenue, Chatham Maritime, Kent ME4 4TB, UK; m.ranatunga@greenwich.ac.uk (M.R.); andrew.deacon@greenwich.ac.uk (A.D.); j.s.boateng@greenwich.ac.uk (J.B.); 2Centre for Health and Life Sciences Research, School of Human Sciences, London Metropolitan University, 166-220 Holloway Road, London N7 8DB, UK; l.harbige@londonmet.ac.uk; 3Halo Labs Ltd., Burlingame, CA 94010, USA; paul.dyer@halolabs.com

**Keywords:** peripheral blood-derived leukocytes, ex vivo, *L. aethiopica^GFP^*, *L. mexicana^GFP^*, CD4^+^, naïve, activated, effector, regulatory T cells

## Abstract

Leishmania parasites are transmitted to mammalian hosts through the bite of sandflies. These parasites can infect phagocytic cells (macrophages, dendritic cells, and neutrophils) and non-phagocytic cells (B cells and fibroblasts). In mice models, the disease development or resolution is linked to T cell responses involving inflammatory cytokines and the activation of macrophages with the M1/M2 phenotype. However, this mechanism does not apply to human infection where a more complex immunological response occurs. The understanding of interactions between immune cells during Leishmania infection in humans is still limited, as current infection models focus on individual cell types or late infection using controlled human infection models (CHIMs). This study investigated the early parasite infection in freshly isolated peripheral blood-derived (PBD) leukocytes over 24 h. Flow cytometer analysis is used in immunophenotyping to identify different subpopulations. The study found that among the *L. aethiopica^GFP^*-associated leukocytes, most cells were neutrophils (55.87% ± 0.09 at 4 h) and monocytes (23.50% ± 0.05% at 24 h). B cells were 12.43% ± 0.10% at 24 h. Additionally, 10–20% of GFP^+^ leukocytes did not belong to the aforementioned cell types, and further investigation revealed their identity as CD4^+^ T cells. Data not only confirm previous findings of Leishmania infection with PBD leukocytes and association with B cells but also suggest that CD4^+^ T cells might influence the early-stage of infection.

## 1. Introduction

Leishmaniasis is a neglected tropical disease that affects more than 20 million people worldwide and is endemic in 98 countries [1]. An estimated 700,000 to 1 million new cases occur annually, with only a small fraction of those infected by parasites causing leishmaniasis eventually developing the disease [2]. Several therapeutic options are available; however, their use is impaired by toxicity, resistance, and high cost of treatment [3,4,5,6]. The disease can also be prevented by reducing exposure to carrier sandflies, using insect repellents, bed nets, and environmental control; however, there is no effective vaccine available for human use. Despite its importance, knowledge about this disease is still incomplete and many challenges remain to be solved. For example, a deep understanding of the immune mechanisms that determine the outcome of infection and the risk of relapse or reinfection is still missing.

One of the ways to advance our understanding of leishmaniasis is to use different models that mimic the infection and the immune response in humans. These models include in vivo, in vitro, and ex vivo studies that have their own advantages and limitations.

In vivo models use animals such as mice, hamsters, and monkeys that are infected with Leishmania parasites and develop clinical signs similar to humans. These models allow the investigation of the natural course of infection, the host-parasite interaction, and the efficacy of drugs and vaccines [7,8]. However, animal models do not fully represent humans, and host-protective mechanisms, complement activation, and professional phagocytes differ. For example, the granulocyte-to-monocyte ratio in human blood is 6–7:1, whereas it ranges from 2–3:1 in mice [9]. Human complement is highly cytotoxic to promastigotes, whereas mouse complement is not [10]. Additionally, certain disease manifestations, such as mucocutaneous leishmaniasis (MCL) and diffuse cutaneous leishmaniasis (DCL), do not develop in mouse or hamster models [11].

Human models are mostly based on studying patients with leishmaniasis. These represent late stages of successful infections and are unsuitable for studying early-stage establishment of infection or asymptomatic infection. Furthermore, environmental changes and inter-individual differences make extrapolation challenging. The recently described controlled human infection model (CHIM) is promising but currently limited for *L. major* infections [12]. CHIM also raises safety and ethical issues, as healthy volunteers are intentionally exposed to pathogens [13].

In vitro models use human and mouse cell lines infected with Leishmania in the laboratory to study the molecular and cellular mechanisms of infection, including parasite invasion, survival, replication, and drug resistance [14,15]. However, these models have limitations, such as the lack of immune cells, loss of cell differentiation, and artificial conditions that do not fully replicate real disease environments. While they only partially represent primary cells, in vitro studies have still significantly advanced our understanding of Leishmania infection [16].

Ex vivo models involve infecting human cells from donor blood with Leishmania parasites in the laboratory to study the immune response of cells like macrophages, dendritic cells, and T cells. Ex vivo studies provide a viable alternative to human, in vivo, and in vitro models, as they are widely accessible and use primary human cells, which may in some cases replicate complex interactions more effectively. Using isolated human peripheral blood mononuclear cells (PBMCs) offers the advantage of employing the same cells found in natural infection, although they lack the intricate network of cell interactions within the blood pool. Furthermore, it can better replicate the natural environment by utilizing blood samples containing all leukocytes, instead of isolating individual cell subpopulations. Despite its advantages, the use of this model has been limited, with only two known papers studying the kinetics of infection in whole blood over 16 h and one paper on genetic studies via immune response on early *Leishmania (Viannia) panamensis* interaction with white blood cells (WBCs) or PBMCs [17,18,19,20].

To better understand leishmaniasis, it is crucial to use a combination of different models to gain a comprehensive view of the disease. This approach helps us identify knowledge gaps and improve diagnosis, treatment, and prevention.

This study aims to use an ex vivo model to investigate the association between each type of peripheral blood-derived (PBD) leukocyte and either *L. aethiopica^GFP^* or *L. mexicana^GFP^* parasites after 4 and 24 h incubation with the freshly isolated mixed population of leukocytes. Individual cell types were identified via flow cytometry following immunophenotyping of neutrophils, monocytes, B cells, and subpopulations of T cells.

## 2. Materials and Methods

### 2.1. Leishmania spp. Promastigotes Cultures

Leishmania parasites expressing genetically modified green fluorescent protein (GFP), namely *L. aethiopica* (MHOM/ET/72/L100) and *L. mexicana* (MNYC/B2/62/M379) [21], were subcultured in Schneider’s Drosophila medium (Invitrogen, Paisley, UK) supplemented with 23% (*w*/*v*) heat-inactivated fetal bovine serum (HI-FBS) (Invitrogen, Paisley, UK) and 1X penicillin/streptomycin/glutamine (Penstrep) (Invitrogen, UK) at a concentration of 5 × 10^5^ cells/mL, at 24 °C.

### 2.2. Isolation of Human Blood Cells via Red Blood Cell Lysis

Approximately 10 mL of peripheral blood was collected from healthy individuals by forearm venipuncture. Blood was transferred into 50 mL centrifuge tubes containing 6000 USP units (United States Pharmacopeia unit) (37.5 mg) of heparin sodium salt (Santa Cruz Biotechnology, Santa Cruz, CA, USA) diluted in 100 µL of 1X phosphate buffer solution (PBS) (Sigma, Hertfordshire, UK). A minimum concentration of 20–140 USP/mL of human blood was maintained to prevent coagulation. The heparinized blood was treated with red blood cell lysis buffer to lyse the red blood cells, following the manufacturer’s instructions (BioLegend, London, UK). Isolate leukocytes were resuspended in complete RPMI-1640 media (Fisher Scientific, Loughborough, UK) supplemented with 10% (*v*/*v*) HI-FBS (Invitrogen, Paisley, UK) and 1X Penstrep (Invitrogen, Paisley, UK), and incubated with Leishmania parasites.

### 2.3. Incubation of Leishmania Parasites with Human PBD Leukocytes

Infective metacyclic promastigotes were isolated using peanut agglutination (50 µg/mL) (PNA, Vector Laboratories, London, UK) from Leishmania in the stationary phase, following the previously described method in Mottram, 2008. Human PBD leukocytes were then incubated with Leishmania parasites at a ratio of 10:1 (Leishmania parasites: human leukocytes) in a 24-well plate. The cells were incubated at 37 °C with 5% CO_2_ for a duration of up to 24 h.

### 2.4. Antibody Staining

The cells were split into two and subjected to staining with two antibody cocktails for a duration of 15 min at 37 °C. The first split stained with a cocktail consisted of CD20, CD14, and CD16b antibodies and identified neutrophils, monocytes, and B-lymphocytes (Figure 1 and Appendix A). The second split stained with cocktail included CD4, CD25, and CD127 antibodies and identified Naïve T cells (CD4^+^, CD25^−^, and CD127^+^), activated T cells (CD4^+^, CD25^+^, and CD127^+^), effector T cells (CD4^+^, CD25^−^, and CD127^−^), and regulatory T cells (Treg) (CD4^+^, CD25^+^, and CD127^−^) (Figure 2 and Appendix A). Subsequently, the cells were washed twice to remove any unbound antibodies and to ensure proper association with the parasites before analysis using an Accuri C6 plus flow cytometer. Staining and data collection were performed after 4 and 24 h of incubation with both Leishmania species.

### 2.5. Flow Cytometry Analysis

During the flow cytometric analysis, red blood cell and debris contamination was excluded by setting a threshold of 10^6^ on the forward scatter height (FSC-H) parameter. The total events were gated as R1 in each sample (Appendix A) and lymphocytes were gated in R2 (Appendix A) with a minimum of >6000 events in R2 per sample for analysis.

Viable cells were gated in the forward scatter versus side scatter (FSC vs. SSC) plot (Appendix A). Cell doublets were removed following analysis of each scatter parameter against its height and area (SSC-H vs. SSC-A and FSC-H vs. FSC-A) (Appendix A).

Parasite-associated cells were identified based on the expression of GFP (count vs. FL1) (Appendix A), and further separation of subpopulations was performed by analysing the membrane stain for each population (Appendix A). The percentage of parasite-associated cells was graphed for both cocktails (Figure 1A,B,D,E in Green GFP^+^ and Figure 2A,B,D,E in Green GFP^+^CD4^+^). Subsequently, further gating was performed within the GFP^+^ cell population to determine the percentage of each Leishmania-interacted cell subpopulation (Figure 1 and Figure 2A,B,D,E in white). The control samples at 4 and 24 h were separated into all subpopulations within the whole population in cocktail 1 (Figure 1C,F) and within the CD4^+^ population in cocktail 2 (Figure 2C,F).

To differentiate cell populations, the following filters were used to differentiate between the following antibodies and associated fluorochromes: PE-Cy7/CD20 for B-lymphocytes, AlexaFluor647/CD14 for monocytes, PE/CD16b for neutrophils, PE-Cy7/CD4, AlexaFluor647/CD127, and PE/CD25 for naïve T cells, activated T cells, effector T cells, and Treg cells (BD Biosciences, Berkshire, UK). These samples were analysed using an Accuri C6 plus flow cytometer.

### 2.6. Statistical Analyses

The results were presented as individual data points for five biological replicates along with the mean ± Standard Error of the Mean (SEM). Statistical comparisons between control and treated groups were analysed using Dunnett’s multiple comparisons test of variance. GraphPad Prism 7 and Rx 64 3.5.1 were used for these analyses. A threshold of *p* ≤ 0.05 was considered significant.

## 3. Results

### 3.1. Leishmania Interaction with Human B Lymphocytes (CD20^+^), Monocytes (CD14^+^), and Neutrophils (CD16b^+^ CD14^−^) over 24 h

The first research question we investigated is whether the *L. aethiopica^GFP^* and *L. mexicana^GFP^* interaction with lymphocytes, monocytes, and neutrophils varies over time and whether this variation is species-specific. The percentage of human cell subpopulations expressing GFP at 4 and 24 h from infection was therefore measured via flow cytometry.

Flow cytometry analysis revealed that 52.83% ± 0.04 of cells were associated with *L*. *aethiopica^GFP^* after 4 h from incubation (Figure 1A in green). No significant (*p* > 0.05) increase was detected after 24 h, with 53.51% ± 0.04 of GFP-expressing cells identified (Figure 1D in green). Interestingly, the *L. mexicana^GFP^* association with leukocytes was lower after 4 h of incubation (40.24% ± 0.09) (Figure 1B in green) and increased significantly (*p* ≤ 0.05) at 24 h from incubation, with 62.15% ± 0.17 of leukocytes expressing GFP (Figure 1E in green).

Further analysis of GFP-expressing leukocytes showed that at 4 h following *L. aethiopica^GFP^* incubation, the leukocyte population consisted of 10.22% ± 0.08 CD20^+^ (B cells), 20.7% ± 0.08 CD14^+^ (monocytes), and 55.87% ± 0.09 CD16b^+^CD14^−^ (neutrophils) (Figure 1A). Similarly, following *L. mexicana^GFP^* association, 14.16% ± 0.11 expressed CD20^+^ (B cells), 25.60% ± 0.07 expressed CD14^+^ (monocytes), and 55.34% ± 0.08 of infected cells expressed CD16b^+^ CD14^−^ (neutrophils) (Figure 1B). Further analysis of GFP-expressing leukocytes showed that at 24 h following *L. aethiopica^GFP^* association, the leukocyte population consisted of 12.43% ± 0.10 CD20^+^ (B cells), 23.50% ± 0.05 CD14^+^ (monocytes), and 59.19% ± 0.08 CD16b^+^CD14^−^ (neutrophils) (Figure 1A).

Similarly, following *L. mexicana^GFP^* incubation, 9.11% ± 0.10 CD20^+^ (B cells), 21.43% ± 0.10 expressed CD14^+^ (monocytes), and 57.37% ± 0.11 of associated cells expressed CD16b^+^CD14^−^ (neutrophils).

In both Leishmania species and at both time points, the CD16b^+^CD14^−^ (neutrophil) population exhibited a significantly higher percentage (*p* ≤ 0.05) compared to CD14^+^ monocytes and CD20^+^ B lymphocytes. Additionally, a significantly higher percentage (*p* ≤ 0.05) of CD14^+^ monocytes was observed compared to CD20^+^ B cells. Notably, the distribution of GFP^+^ subpopulations differed from the distribution of uninfected control leukocyte subpopulations (Figure 1C,F).

The control group consisted of leukocytes incubated under identical conditions as cells exposed to parasites but without Leishmania parasites. After 4 h, the control samples exhibited a higher proportion of monocytes (~25%) compared to neutrophils (~15%) and B cells (~15%), while at 24 h, the proportion was dominated by neutrophils (~35%), followed by monocytes (~15%) and B cells (~10%). These findings indicate that at 4 h, a greater percentage of parasites were bound to neutrophils than to monocytes, regardless of the relative abundance of different cell types.

When examining the subpopulations of GFP^+^ leukocytes, B cells, neutrophils, and monocytes accounted for approximately 86% to 95% of the Leishmania-associated cells. This suggests that other subpopulations, such as T cells, may have the potential to support Leishmania interaction. To investigate this possibility, specific staining for T lymphocytes was conducted.

### 3.2. Leishmania Interaction with Subpopulations of Human T Lymphocyte (CD4^+^) over 24 h

The second research question was whether Leishmania parasites directly interact with human T-lymphocytes, and if this interaction varies in time and depending on the infecting species.

Flow cytometry analysis of cells incubated with *L. aethiopica^GFP^* revealed that 10.00% ± 0.02 of GFP^+^ cells expressed CD4^+^ at 4 h (Figure 2A in green) and 23.31% ± 0.05 of GFP^+^ cells expressed CD4^+^ at 24 h (Figure 2D in green). Of the GFP-expressing CD4^+^ population, at 4 and 24 h respectively, 16.12% ± 0.11 and 23.31% ± 0.05 were naïve cells, 52.99% ± 0.12 and 67.77% ± 0.08 were activated cells, 4.62% ± 0.03 and 4.91% ± 0.07 were effector cells, and 26.26% ± 0.05 and 19.09 ± 0.07 s were Treg cells, respectively.

During *L. mexicana^GFP^* incubation, 19.37 ± 0.06 of GFP^+^ cells expressed CD4^+^ at 4 h (Figure 2B in green) and 11.88% ± 0.06 of GFP^+^ cells were CD4^+^ at 24 h (Figure 2E in green) within the GFP^+^ populations. Of the GFP-expressing CD4^+^ population, at 4 and 24 h, 27.68% ± 0.15 and 7.49% ± 0.04 were naïve cells, 29.93% ± 0.18 and 70.62% ± 0.2 were activated cells, 9.53% ± 0.06 and 1.26% ± 0.004 were effector cells, and 29.29% ± 0.21 and 20.63 ± 0.16 were Treg cells, respectively.

Interestingly, the percentage of activated T cells was significantly higher at 24 h for both Leishmania species and at 4 h for *L. aethiopica^GFP^*, compared to other subpopulations (Figure 2A,D,E). Although the other subpopulations contributed in a significantly lower percentage to the GFP+ population, both *L. aethiopica^GFP^* and *L. mexicana^GFP^* parasites were capable of interacting with all four CD4+ subpopulations at both time points.

These findings indicate that 5–10% of the associated population not accounted for by cocktail one (CD20, CD14, and CD16b) contains CD4^+^ cells.

## 4. Discussion

These findings indicate that both *L. aethiopica^GFP^* and *L. mexicana^GFP^* are associated with neutrophils and monocytes. This observation aligns with a previous study by Moreno et al. [18], which demonstrated that *L. donovani* and *L. amazonensis* infect both cell types ex vivo. Interestingly, a significantly higher proportion of neutrophils was associated with both *L. aethiopica^GFP^* and *L. mexicana^GFP^* compared to monocytes (*p* ≤ 0.05). The preferential association of neutrophils by *L. aethiopica^GFP^* and *L. mexicana^GFP^* metacyclic promastigotes, despite their comparable abundance with monocytes and B cells, indicates a greater tropism of the parasites for this cell type and highlights a significant interaction. Despite the well-known role of monocytes/macrophages in supporting Leishmania association, this finding is consistent with in vivo investigations that have reported neutrophils as the initial target cells for infection [22,23,24,25].

Neutrophils are known to serve as host cells for *L. infantum* and *L. braziliensis* in mice [26,27], as well as in ex vivo investigations using purified human neutrophils with *L. major* [28] and *L. aethiopica* [29]. 

Consistent percentages, i.e., approximately 11%, of infected neutrophils have been observed after 3 h of exposure to *L. guyanensis* and *L. shawi*, *L. infantum*, and *L. amazonensis* in mice [30]. In contrast, the findings presented in this study indicate that the percentage of associated neutrophils is comparable between *L. aethiopica^GFP^* and *L. mexicana^GFP^* at both 4 and 24 h post-incubation. The proportion of infected neutrophils appears, therefore, to vary depending on the infecting Leishmania species and the model system being studied.

The entry of parasites into neutrophils involves the recognition of pathogen-associated molecular patterns (PAMPs) by pattern recognition receptors (PRRs), including Toll-like receptors (TLRs) and NOD-like receptors (NLRs) [22]. The exact role of neutrophils in Leishmania infection remains unclear. Neutrophils may support the establishment of infection and clinical disease by acting as “Trojan horses” or “Trojan rabbits” [31]. They harbour parasites and become apoptotic neutrophils, which play a dual role. On one hand, they facilitate the establishment of parasites within macrophages, promoting the resolution of inflammation and providing protection to the parasites. This is achieved through the release of anti-inflammatory molecules such as transforming growth factor beta (TGF-β1) and Prostaglandin E2 (PGE_2_) [32,33,34]. On the other hand, neutrophils contribute to the elimination of Leishmania parasites through the induction of neutrophil extracellular traps (NETs) and the release of antimicrobial components [35]. The precise role of human neutrophils in Leishmania infection is not fully understood, but studies suggest that the outcome depends on the balance between these opposing actions, with factors such as TGF-β1 and PGE_2_ influencing the parasite burden and infection outcomes in macrophages [36].

As expected, the presented data demonstrate that monocytes can support association with Leishmania parasites over time. Approximately 20% of the infected cells identified during ex vivo incubation were identified as monocytes, regardless of the Leishmania species or time points. Macrophages serve as the primary host cells for Leishmania, facilitating parasite differentiation, replication, and spread. Parasites can enter macrophages either directly or via neutrophils [36]. In our model, the percentage of infected neutrophils remained unchanged, indicating that parasites enter monocytes directly rather than through neutrophils. The entry of parasites into macrophages is well characterized and involves various receptors such as complement receptors (CRs), mannose receptors (MR), fibronectin receptors, and Fcγ receptors (FcγRs) [37,38,39,40].

B-1 cells obtained from the peritoneal cavities of BALB/c mice possess phagocytic capabilities and internalize *L. major* and *L. amazonensis* parasites [20,41]. Interestingly, in the ex vivo model utilized by the authors, it was not observed that B cells exhibited significantly higher infection rates compared to peritoneal macrophages and medullary macrophages during in vitro studies conducted at 16 and 24 h post-infection [20]. However, it should be noted that the antibody employed to identify B cells was CD19, which is considered a less specific marker for the B cell lineage compared to CD20 [42].

Although it cannot be fully concluded that the presence of GFP in the leukocyte population is representative of association, the findings presented in this study are consistent with the fact that *L. aethiopica^GFP^* and *L. mexicana^GFP^*, similar to *L. amazonensis* and *L. major*, might enter B-1 cells. Although the percentage of GFP-expressing B cells was significantly lower than monocytes and neutrophils, it remained consistent for both species at both time points. Therefore, these results support previous research indicating that isolated B cells can serve as hosts for Leishmania parasites, even in the presence of other types of leukocytes [20]. In the context of whole-blood incubation, no species-specific differences in the association of neutrophils, monocytes, and B cells were observed between *L. aethiopica^GFP^* and *L. mexicana^GFP^*.

After 4 h and 24 h of incubation with both species, approximately 10% of the blood leukocytes associated with parasites did not belong to the populations of neutrophils, monocytes, or B cells. Further analysis revealed that these missing cells were CD4^+^ T cells. This study and an unpublished study carried out by us in immortalized human T cells, i.e., Jurkat cells (Harbige, Deacon, Getti), therefore, represents the first evidence demonstrating the association of human T cells with infecting Leishmania parasites. The findings presented in the present study also suggest species-specific behavior differences following CD4^+^ T helper cell incubation with Leishmania. The percentage of GFP-expressing CD4^+^ T helper cells is significantly higher following 4 h of incubation with *L. mexicana^GFP^* than *L. aethiopica^GFP^*. Interestingly, the opposite pattern was observed following 24 h of incubation, where *L. aethiopica^GFP^* association significantly increased while *L. mexicana^GFP^* association decreased. These results suggest that *L. mexicana^GFP^* has a higher affinity for binding to CD4^+^ T helper cells at 4 h, indicating that over time, CD4^+^ T helper cells may be capable of eliminating *L. mexicana^GFP^* while continuing to support the persistence of *L. aethiopica^GFP^*.

In the control group, naïve T cells accounted for 50% of the CD4^+^ cell population. However, at 24 h, only 8% of the CD4^+^ population consisted of naïve T cells. Interestingly, a lower percentage of activated T cells (~30%) was observed in the control sample, but a significantly higher percentage of activated T cells was detected within the parasites associated with CD4^+^ populations. This suggests that naïve T cells may undergo differentiation into activated T cells upon attachment during Leishmania inoculation. With the exception of the association with *L. mexicana^GFP^* (~30%) at 4 h, activated T cells represented a significantly higher percentage of association (~53–72%) with *L. aethiopica^GFP^* at both time points and *L. mexicana^GFP^* at 24 h, which implies that the infected naïve cells differentiate into activated cells.

In the control samples, a small percentage of effector T cells was observed, and as anticipated, in the samples incubated with parasites, a low percentage of association was detected in both species at both time points. Effector T cells are known to produce elevated levels of the pro-inflammatory cytokine Interferon-gamma (IFN-γ), which facilitates parasite elimination and contributes to protective immunity during Leishmania infection [42]. Therefore, the involvement of effector T cells in the immune response against Leishmania is crucial. However, whether parasites have the ability to infect effector T cells and potentially influence the inflammatory response or modulate cytokine production remains to be established and warrants further investigation.

Treg cells were associated with Leishmania parasites during the PBD leukocyte incubation at both time points. Although the control samples exhibited a cell population of approximately 10% at both time points, a higher percentage of associated Treg cells, ranging from 20% to 30%, was observed in both species at both time points. Treg cells are known to produce Interleukin 10 (IL-10) and TGF-β, which exert species-specific influences on disease outcomes. In the case of *L. donovani* and *L. braziliensis* infections, Treg cells contribute to disease control, whereas in *L. amazonensis* infection, Treg activation benefits the survival of parasites and the maintenance of the memory response [43,44,45]. However, the susceptibility of T cells to Leishmania association has not been previously investigated, and the data presented in this study demonstrate that T cells can support the presence and survival of Leishmania when bound for a period of 24 h.

This study provides evidence that *L. aethiopica^GFP^* and *L. mexicana^GFP^* can infiltrate distinct subpopulations of white blood cells, including B cells, neutrophils, and macrophages, during human-derived whole blood infection. Additional investigations are still required to determine whether T cells establish a transient or persistent association as host cells for Leishmania and how this interaction might impact the immune response equilibrium. The association of Leishmania with T cells is likely to modulate the immune response. Infected T cells may produce cytokines like IFN-γ and TNF-α, which are crucial for activating macrophages to kill the parasites. However, this can also lead to excessive inflammation and tissue damage [46]. If Leishmania can persist in T cells, leading to chronic infection, this could result in long-term immune activation and potential tissue damage [47]. The potentially important clinical impact of identifying specific subsets of T cells that can support Leishmania infection could lie in the elucidation of a more complete understanding of the evasion mechanisms employed by these parasites. This would be beneficial in identifying biomarkers such as specific cytokine profiles or memory T cell populations that can improve the diagnosis and monitoring of leishmaniasis and the development of new effective treatments and vaccines.

## Figures and Tables

**Figure 1 microorganisms-12-01909-f001:**
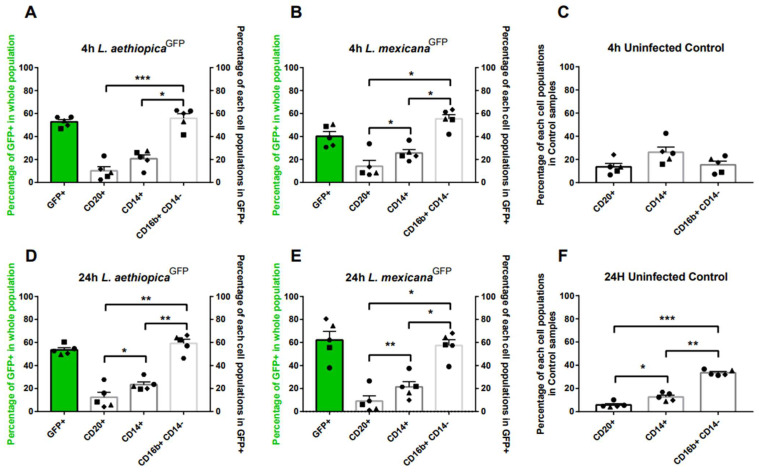
The percentages of the GFP^+^ population in human B lymphocyte (CD20^+^), monocyte (CD14^+^), and neutrophils (CD16b^+^ CD14^−^) were determined over 24 h. Human whole blood was lysed and incubated with GFP-expressing metacyclic promastigotes over 24 h. ((**A**,**B**,**D**,**E**) in green). The percentage of GFP^+^ cells in the whole population was detected via the Accuri C6 plus flow cytometer and FL1 channel in the gated whole population ((**A**,**B**,**D**,**E**) in white). Within the GFP+ population, further analysis was carried out to determine the percentage of each subpopulation; B cells (identified as CD20^+^ lymphocytes), macrophages (identified as CD14^+^), and neutrophils (identified as CD16b^+^ CD14^−^) were separated. Control samples (**C**,**F**) at 4 h and 24 h were separated into all three populations of B cells (identified as CD20^+^ lymphocytes), macrophages (identified as CD14^+^), and neutrophils (identified as CD16b^+^ CD14^−^). Statistical analysis was performed by Dunnett’s multiple comparisons, plotted bars correlate to the standard error as follows: *, *p* ≤ 0.05; **, *p* ≤ 0.01; *** and *p* ≤ 0.001.

**Figure 2 microorganisms-12-01909-f002:**
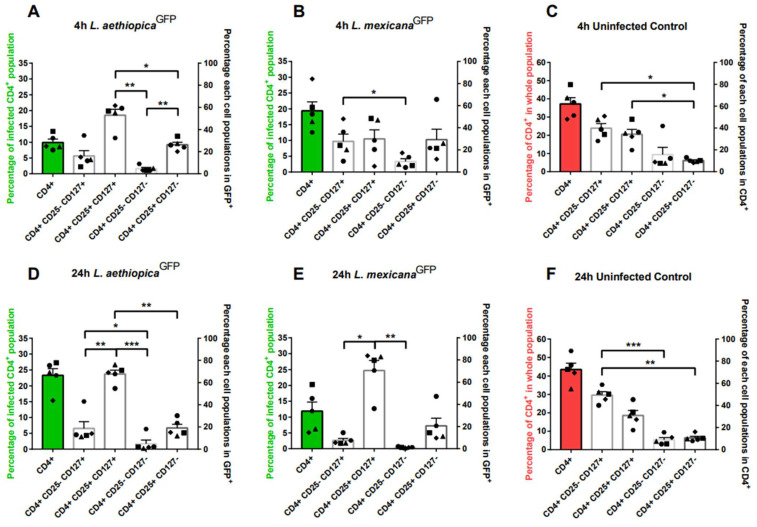
The impact on the proportion of the distinct T cell subpopulations within the GFP^+^ CD4^+^ population over 24 h. Subpopulations were identified based on specific cell surface markers; naïve T cells (identified as CD4^+^, CD25^−^, and CD127^+^), activated T cells (identified as CD4^+^, CD25^+^, and CD127^+^), effector T cells (identified as CD4^+^, CD25^−^, and CD127^−^) and T_reg_ (identified as CD4^+^, CD25^+^, and CD127^−^) via a Accuri C6 plus flow cytometer. Flow cytometry analysis was employed to detect the percentage of incubated cells within the entire population, using the FL1 channel within the gated population. Subsequently, within the GFP^+^ population, the percentage of each CD4^+^ subpopulation was determined ((**A**,**B**,**D**,**E**) left y-axis in green). Within that CD4^+^ population, the percentage of subpopulations was separated at 4 and 24 h time points ((**A**,**B**,**D**,**E**) right y-axis). (**C**,**F**) control samples at 4 and 24 h were separated into all four subpopulations within the CD4^+^ population. Statistical analysis was performed by Dunnett’s multiple comparisons, plotted bars correlate to the standard error as follows: *, *p* ≤ 0.05; **, *p* ≤ 0.01; and ***, *p* ≤ 0.001.

## Data Availability

The original contributions presented in the study are included in the article/Appendix A, further inquiries can be directed to the corresponding author/s.

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
