# Peer review of "Ex Vivo Analysis of the Association of GFP-Expressing *L. aethiopica* and *L. mexicana* with Human Peripheral Blood-Derived (PBD) Leukocytes over 24 Hours"

_microorganisms, 2024, doi:10.3390/microorganisms12091909_

Round 1

Reviewer 1 Report

Comments and Suggestions for Authors

This manuscript has reference value for revealing the pathogenic mechanism of Leishmania, but there are some shortcomings.

1 . Background needs refining.

2. The description of the results should be revised to be more clear, and it is recommended to use the table in moderation.

3.The research methods used are relatively simple.

4. Leishmania parasites, of genetically modified green fluorescent protein (GFP) expressing, L. aethiopica (MHOM/ET/72/L100) and L. mexicana (MNYC/B2/62/M379).The origin of these strains is not stated, and the researchers themselves constructed them?

Comments on the Quality of English Language

/

Author Response

Please see the attached revision for reviewer 1. 

Reviewer 2 Report

Comments and Suggestions for Authors

The paper entitled "Ex vivo analysis of GFP-expressing L. aethiopica and L. mexicana association with human blood-derived (PBD) leukocytes over 24 hours," studied by Rantunga et al., offers valuable insights into the early immune response to Leishmania infection. By examining interactions in freshly isolated PBD leukocytes during the first 24 hours of infection, the study reveals through flow cytometry that neutrophils (59.19%) are the primary cell type associated with L. aethiopica at 4 hours, with a notable increase in monocytes (23.50%) and B cells (12.43%) by 24 hours. Interestingly, CD4+ T cells were also identified among the infected leukocytes, suggesting an early role in the immune response and challenging existing models that focus on individual cell types or later stages of infection. While the study effectively underscores the importance of these early cellular interactions, it could benefit from an extended observation period and functional analyses to fully elucidate the contributions of these immune cells. Overall, the manuscript is well-prepared for publication in this journal, but I recommend some modifications before submission. The introduction is overly long and should be condensed. When describing the results, it would be clearer to first state the research question, followed by methods, results, and finally the conclusions drawn from the findings, rather than just presenting percentages without context.

Author Response

Please see the attached revision for reviewer 2. 

Reviewer 3 Report

Comments and Suggestions for Authors

Overall, the study is well-designed and the manuscript is well written. I just have two minor suggestions.

1. Please shorten the introduction. The present form is too lengthy.

2. In the discussion, please add the clinical significance and discuss the clinical implication based on the findings of the present work.

Author Response

Please see the attached revision for reviewer 3. 

Round 2

Reviewer 1 Report

Comments and Suggestions for Authors

The author has made improvements and answers to the review comments, but there are still small problems, 

1 .such as the format of references carefully modified according to the format of this journal.

2. Check numbers and statistical analysis.

Comments on the Quality of English Language

/
